# A Framework for Interfacing and Partnering with Environmental Justice Communities as a Prelude to Human Health and Hazard Identification in the Vulnerable Census Tracts of Columbus, Ohio

**DOI:** 10.3390/ijerph192113846

**Published:** 2022-10-25

**Authors:** Heather Lochotzki, Karen Patricia Williams, Cynthia G. Colen, Olorunfemi Adetona, Charleta B. Tavares, Georgina M. Ginn, Rejeana Haynes, Wansoo Im, Tanya Bils, Darryl B. Hood

**Affiliations:** 1Division of Environmental Health Sciences, College of Public Health, Ohio State University, Columbus, OH 43210, USA; 2Martha S. Pitzer Center for Women, Children & Youth, College of Nursing, The Ohio State University, Columbus, OH 43210, USA; 3Department of Sociology, The Ohio State University, Columbus, OH 43210, USA; 4PrimaryOne Health, 2780 Airport Drive, Suite 100, Columbus, OH 43219, USA; 5Columbus Early Learning Centers, 1611 Old Leonard Avenue, Columbus, OH 43219, USA; 6St. Vincent Family Services, 1490 East Main Street, Columbus, OH 43205, USA; 7Department of Family and Community Medicine, Meharry Medical College, Nashville, TN 37208, USA

**Keywords:** community-based participatory research (CBPR), community engagement, environmental justice, hazard identification, *Public Health Exposome*, stakeholder coalition, urban census tracts, environmental public health practitioner

## Abstract

Columbus, Ohio is one of the more prosperous, well-educated, and progressive cities in the United States. However, it ranks as the second worst life expectancy at birth, has a census tract wealth gap (27-year disparity), and one of the higher infant mortality rates in the country. These data suggest that there are likely several high-risk, vulnerable neighborhoods in Columbus with residents experiencing disparate and adverse outcomes. Illustrative of this fact are studies that have examined the social processes and mechanisms through which neighborhood contexts are at the forefront, including exposures to chemical stressors such as particulate matter (PM_2.5_) as well as non-chemical stressors including violence, social determinants of health, zoning, and land use policies. It is documented that disparate and adverse outcomes are magnified in the vulnerable neighborhoods on the Near East Side as compared to Columbus city proper, Franklin County and/or the state of Ohio. As such, we developed a nuanced community engagement framework to identify potential environmental hazards associated with adverse pregnancy outcomes in those census tracts. The refined framework uses a blended version of traditional community-based participatory research (CBPR) models and is referred to as E^6^, Enhancing Environmental Endeavors via e-Equity, Education, and Empowerment.

## 1. Introduction

The purpose of the E^6^ model is to bring comprehensive primary care and health care services to where people live, work, play and pray. Over the past 50 years, this has proved to be easier said than done in the vulnerable census tracts of the United States. Our efforts to formalize and create a template for creation of a functional, interdisciplinary, community-based research stakeholder team in true partnership with residents is transformative and will positively impact individual, community, and population health. This model can assist with informing the decision-making processes related to resource allocation for high-risk and vulnerable communities by local and state environmental public health policy officials. The development of the E^6^ framework for implementation in vulnerable communities was to bring awareness and to educate residents in these census tracts of the potential hazards from chemical and non-chemical stressor environmental exposures in a series of in-person community meetings (Figure 1).

This was accomplished by having open forum discussions with residents of the King-Lincoln district, the Near East Side, and Mt. Vernon communities of Columbus, Ohio. These meetings entailed extended discussions that surveyed potential environmental hazards near where these residents live, work, play, and pray. In 2017 we developed a public participatory geographical information system (PPGIS) as a tool to cultivate general awareness and educational literacy among the residents regarding the issue of environmental hazards (https://www.immappler.com/osudehs/ (accessed on 26 August 2022)) [1,2,3]. The literature shows that citizen science and the engagement of residents in data collection is invaluable to building a collaborative effort between scientists and community members to address environmental concerns [4]. The critical component in the E^6^ framework that has emerged as the rate limiting factor, is the ability to engage partners that have a strategic interest in promoting positive outcomes within these communities. Over a 4-year time period that spans the pre-COVID-19 syndemic period of August 2018 through September 2022, we were able to enhance our pre-existing partnerships with our stakeholders: CareSource of Ohio, Primary One Health, Columbus Early Learning Centers, and St. Vincent Family Center, with The Ohio State University Colleges of Public Health and Nursing. Formal engagement with residents occurred at a series of three community meetings that took place in November 2019, January 2020, and February 2020 at the Ohio State University African American and African Studies Community Extension Center and at the Columbus Early Learning Center located in the Mt. Vernon community.

The expected outcome of these meetings was to create a cohesive plan for establishing a demonstration project that would characterize and focus on the “at risk” census tracts with adverse pregnancy outcomes (pre-term birth, low birth weight and high infant mortality) [5]. Cifuentes et al., (2019) is the publication that served as the scientific premise for our continued involvement with the Near East Side. This publication sought to understand the association of several environmental and socio-demographic variables with adverse pregnancy outcomes across the 88 counties in Ohio by modelling an African American woman’s cohort in Ohio. The adverse outcomes queried were pre-term birth (PTB) and low birth weight (LBW) as a proxy for infant mortality. The proposed project would bring resources, comprehensive primary care, and other health care services directly to where residents live, work, play, and pray. Our E^6^ model, shown in Figure 1 above, was viewed by residents as an environmental public health care delivery model that would solidify a functional, interdisciplinary, community-based research stakeholder team to be in true partnership with community residents. The conclusion of the community meeting series coincided with the nation going into lockdown due to the COVID-19 syndemic. As a result, for the remainder of 2020, 2021, and the first quarter of 2022, all materials and data were organized and analyzed. A grant proposal was then prepared and submitted to query many of the questions and concerns presented by the community residents.

While Columbus, Ohio is one of the more prosperous, progressive, and well-educated cities in the United States, it exhibits a 27-year disparity in life expectancy at birth per census tract, which ranks the city as the second worst by metro area in the United States (falling just behind Washington, D.C.) [6]. This disparity is associated with the substantial wealth gap, which is among the widest in the United States [6]. This wealth gap suggests that there are vulnerable communities in Columbus whose residents lack adequate access to healthcare and resources and likely face adverse exposures to chemical, such as particulate matter (PM_2.5_), and to non-chemical stressors, such as low socioeconomic status, substandard housing, and neighborhood violence [7]. As previously mentioned, Columbus has one of the highest infant mortality rates in the country. Other adverse pregnancy outcomes that occur in association with infant mortality have been documented in vulnerable communities in Ohio and are shown in Table 1. Vulnerable communities are faced with a plethora of environmental issues that contribute to the observed disparate health outcomes [1]. It is anticipated that the current COVID-19 syndemic exacerbates the known historical disparate health outcomes that are faced by residents living in these communities [7,8,9] The demographic profiles of vulnerable residents in Columbus, Ohio communities prior to the COVID-19 syndemic indicated that they already faced adverse health outcomes as compared to Franklin County, Columbus, or the state of Ohio (Table 1). Residents of the Near East Side and Mt. Vernon communities (zip codes 43203 and 43205) are predominately African American [10]. We have observed across the United States that the impacts of the COVID-19 syndemic are exacerbated in predominately African American communities, due to the disproportionate levels of poverty, social determinants of health, infant mortality, and incarceration [11,12]. Exposures to chemical and non-chemical stressors are thought to adversely impact allostatic load and contribute to the disparity observed among black and white COVID-19 mortality rates [13]. Meanwhile, we know that while black patients exhibit a higher disease burden, they are woefully underrepresented in the literature. Adequate race and ethnicity data is not uniformly or consistently collected and reported in COVID-19 research studies [14]. Illustrative of this fact is a study modeling reading development data among kindergarten children during the current post-COVID-19 syndemic phase, where the larger percentage of the sample population was white children [15].

To address the schema among the vulnerable populations living in environmental justice communities, we developed a refined, innovative engagement model, referred to as E^6^, Enhancing Environmental Endeavors via e-Equity, Education and Empowerment. This model utilizes a functional, multidisciplinary, community-based research stakeholder team to educate and improve the quality of life for residents living in environmental justice communities, as well as to foster citizen science among residents and researchers. The E^6^ model prioritizes the perspectives and desires of the residents and is a novel approach to bring effective mitigation strategies directly to where residents live, work, play, and pray. This method prioritizes accountability and transparency of the stakeholder team through their visibility and presence in the communities in which they are involved. The E^6^ model utilizes our previously described *Public Health Exposome* database and framework [1,5,16], along with Big Data to Knowledge analytics to quantify the outcomes in the environmental justice communities. In terms of methodology, the E^6^ model evolved over 5-phases, as shown below in Figure 2. Application of this framework will contribute to the advocacy and literature deficits pertaining to the empirical data that is necessary to begin the process of addressing policy at the local levels in environmental justice communities. In this regard, there is a growing literature which supports models that demonstrate collaboration of community partners toward enacting policy change [17,18,19,20].

## 2. Materials and Methods

### 2.1. Phase 1

The Ohio State University affiliated scientists did not involve themselves in the Stambaugh-Elwood, the Near East Side, and Milo Grogan communities with the assumption that they knew what the residents wanted and needed to improve environmental and health outcomes. Community leaders approached the Ohio State University environmental public health team with a collaboration proposal to improve the health and well-being of residents living in the Stambaugh Elwood neighborhood [1]. Through meetings with health advisory committees and community leaders, the Ohio State University environmental public heath team learned of the environmental and public health concerns residents had regarding where they lived, worked, played, and prayed. An Environmental Exposure Questionnaire was developed to assess the environmental exposure perspectives of community residents [21] (see Appendix A). Existing community leaders served as the liaison between residents and public health practitioners to foster trust and open communication. This served as the initial template for establishing a multi-stakeholder team that would be capable and poised to deliver the healthcare, resources, and deliverables that were requested and needed by the residents. The community leaders and the Ohio State University team decided that a community engaged approach would be appropriate for this project and would be most beneficial to community residents. The Ohio State University team felt that it was obligatory to meet the residents and extend an invitation to the residents to come together for a series of open meetings.

### 2.2. Phases 2 and 3

Upon receipt of the community leaders’ proposal to the Ohio State University College of Public Health researchers, the team sought to establish a community advisory board. This board would preside over three community meetings toward solidification of a multi-stakeholder research team to address concerns of community residents. With letters of support from each member of the team, the Ohio State University College of Public Health partnered with the Near East Side residents, Ohio State University College of Nursing (Pitzer Center for Women, Children and Youth), Primary One Health (FQHC provider), The Ohio State University College of Medicine (Family Medicine and OB-GYN mobile health unit), Columbus Early Learning Centers (K4 educational learning & development), CareSource of Ohio (Medicare provider), and St. Vincent Family Center (Family health services). Community leaders with the multistakeholder research team collaborated to plan and organize the details and logistics of three community meetings that were to be held in three different locations within the Near East Side and King-Lincoln District of Columbus, Ohio. Details of the meeting logistics were disseminated to the community residents via paper flyers, social media, and word of mouth.

### 2.3. Phase 4

The community meetings were held on 2 November 2019, 16 January 2020, and 20 February 2020, with approximately 50–60 individuals in attendance at each meeting. The multistakeholder team was focused on “developing a collective efficacy” to bring information, potential solutions, and mitigation strategies to residents. Residents were encouraged and incentivized to attend each community meeting by receiving a light dinner and refreshments prior to the meeting and a $25.00 gift card. There was an agenda for each meeting that was driven by the community. However, the meetings were conducted in an open forum format to allow residents to directly voice any concerns and opinions and provide them a platform to engage in brainstorming and conversations with the research team in a face-to-face manner. Through this approach, the collective community stakeholder team was synchronized in their investment in the residents and their proposed concerns. Our customized PPGIS portal built on the MapplerX platform was used as a communication portal for active exchange between residents and the stakeholder team to kick off each community meeting. The PPGIS portal was also utilized as an environmental education tool to share environmental literacy resources and public available data between the team and residents. Using a PowerPoint presentation, it was explained to residents how to access the PPGIS portal on their mobile phones. This methodology promotes the citizen science approach that further forges strong relationships between residents and a functional, multidisciplinary stakeholder team.

## 3. Results and Discussion from Community Meetings

2 November 2019. From this first community meeting, an outline of concerns and suggestions regarding disparate health outcomes in the King-Lincoln District of Columbus, Ohio proposed by the community residents was created. The chemical and non-chemical stressor exposures from the residents’ built, natural, physical, and social environments were considered. Conversations among the residents and panelists focused on the history and current situation of the King-Lincoln District. The South Side cancer cluster and specific community members impacted by it were among the prominent discussions. This included the residents’ call for compensation for the losses and harm they have incurred from the various environmental hazards present in their neighborhoods. In response to this call, the panelists urged and voiced support for a collective community action plan and the utilization of research and data to confront policymakers and ultimately impact policy change. The panelists emphasized that class action lawsuits are expensive and that it may be difficult to procure the necessary legal support. The panelists instead recommended the National Association for the Advancement of Colored People (NAACP) action funding. Gentrification was another passionate topic discussed at the meeting. The panelists discussed that a more modern and positive term may replace “gentrification”, as it has racist origins. The panelists emphasized that racial health and wealth disparities persist. Thus, race and socioeconomic status do matter and must be included in the conversations regarding neighborhood concerns.

16 January 2020. At this meeting, there was a conversational focus on the high infant mortality, low birth weight, preterm birth, and teenage pregnancy rates, along with the disparate health outcomes of those living in the Columbus, Ohio zip codes of 43205, 43207, and 43209. Dr. Hood lead discussions on how these issues are studied in a public health research setting. He explained the *Public Health Exposome* framework, Bayesian network analysis, and the characterization of how exposure to various chemical and non-chemical stressors might influence residents’ susceptibility to disease. Associations between socio-demographic and environmental variables and an adverse health or otherwise negative outcome were analyzed, as reviewed in Cifuentes et al., (2019). These associations accounted for 32.85% density and an average degree of 9.2. Post hoc values of arrows (associations) were plotted as *p*-values based on linear conditional correlation and line widths were highest for the lowest *p*-values. Automatic visualization accounted for the relative value of the links, which was obtained by transforming *p*-values by log-transformation and normalization/truncation from 5 to 1 by a mapping algorithm. This was followed by an energy-based algorithm, available in Pajek software, which located more connected nodes in the center of the graph. The *p*-values for each association were calculated to derive an adjacency matrix representing significant associations between pairs of environmental and/or socio-demographic variables that are controlled by all remaining variables. The weighted adjacency matrix of the resulting Bayesian network model contained all significant *p*-values (<0.05) for each link [5].

The community residents raised their concerns regarding the preservation of community culture, procuring explanations of academic research that are digestible for residents, drug addiction, and crime. Residents emphasized the need for accessible and widespread dissemination to community residents of educational materials on public health topics, such as infant mortality, teenage pregnancy, and opioid addiction. The residents in attendance explained to the panelists that they desired an accountability component for the organizations and stakeholders with a presence in the community. The responses from the Ohio State team included the importance of bringing healthcare and resources directly to the communities that need them, collaboration amongst the stakeholder team to provide funding opportunities for the community, and the commitment to bringing a demonstration project to this adversely impacted community.

20 February 2020. The discussions held at this community meeting carried over from the previous meeting to focus on the prevalent health disparities of residents living in the Mount Vernon neighborhood of Columbus, Ohio. The higher rates of infant mortality, preterm birth, depression, cardiovascular disease, stroke, and diabetes present in this neighborhood compared to Columbus, Franklin County, and the state of Ohio, were discussed. Priorities presented at this meeting by the residents included the employment of culturally competent healthcare clinical staff, mobile clinics providing lactation support and children’s school physicals, healthcare services for the uninsured, family services, childcare, and bringing a community emphasis to African American male healthcare services, such as prostate cancer testing. The residents expressed their satisfaction with current mobile health services provided by other health care providers and expressed interest in more of these types of services coming to their community on a regular schedule. The residents also voiced concerns regarding making healthcare services accessible to community members with disabilities. CareSource emphasized the organization’s commitment to this by echoing the need for more mobile healthcare services in this area of the city. The overall discussion emphasized the priority for mental health services, family services, contraceptive healthcare services, and nutrition resources to be brought to the community residents.

### Phase 5

It has been established in the literature that where an individual lives, works, plays, and prays influences their potential exposure to chemical and non-chemical hazards [22]. Exposure to environmental contaminants, hazards, and to air pollution is not a “random” occurrence across the United States. Race and class play a major role in the historic and current location and clustering of point source emitters, landfills, and waste incinerators [23,24,25,26,27,28,29,30,31]. Exposure to these environmental contaminants has disparate health impacts on residents of low-income and minority communities [32]. Historical and current data shows that the zip codes with higher levels of air pollution and hazardous waste activity are home to predominately residents of color and those with higher poverty rates [22,33]. Health disparities, such as low birth weight, premature birth, infant mortality, cardiovascular disease, diabetes, chronic diseases, and depression have become public health crises in these communities in concordance with such environmental exposures [32].

Community-based participatory research has been a method of interest for engaging and building trust among community residents and scientists. It addresses the disparate health outcomes prominent in vulnerable populations, as it prioritizes the social determinants of health at the neighborhood level, rather than solely at the individual level [34]. Racial and ethnic minority populations experience disproportionally higher rates of disease when compared to their white counterparts, which have only been exacerbated during the COVID-19 syndemic period [35]. These community-based methods can foster resilience in environmental justice communities [36]. However, maintaining community engagement when implementing community-based participatory research methods can present challenges for public health practitioners and community engaged scientists [37]. It has been shown that full community engagement is effective and necessary to observe measurable improvements in health and environmental disparities, yet the actual implementation of such models is still not widely and effectively utilized [38,39]. As these community engagement models are innovative, there has been limited development and consensus as to how to quantify the mechanisms, refinement, and outcomes of such theoretical models [40]. In this systematic review (Figure 3), we review the literature in which community-based participatory research or community engagement is utilized to address environmental health concerns of residents living in environmental justice communities in the United States. We queried the online archive of journal articles in the Scopus, Embase, and PubMed databases. Of the 480 articles imported for screening, 163 duplicates were removed. Of the 317 articles screened, 105 articles were assessed for eligibility. Of those 105 articles, 80 articles were excluded for being the wrong subject matter, wrong document format (review, commentary, multi-case study), or wrong setting. The remaining studies implemented community-based participatory research models in an environmental justice community in the United States. These studies were reviewed using five dimensions: (1) Environmental concerns addressed; (2) Impact populations; (3) Methodologies; (4) Outcomes; and (5) Limitations/Challenges.

Dimension 1: Environmental concerns addressed. There is a myriad of environmental concerns that may warrant researchers to implement community-based participatory research methodologies as tools for hazard mitigation and for addressing disparate health outcomes at the community-level. Concerns include addressing mosquito populations that result from structural variation in the decay of urban housing [41], volatile organic compounds (VOCs) exposure from industrial and mobile sources [42], a chemical disaster [43], industrial pollution [44], fracking [45], contamination of drinking and surface water supplies [46], and the sustainability of watersheds [47].

Dimension 2: Impact populations. The implementation of community-based participatory research models is often conducted in vulnerable communities that are impacted by disparately high levels of chemical and non-chemical environmental stressors. These communities tend to consist predominately of minority populations and those of lower socioeconomic status. A citizen science approach can be taken to conduct ambient air monitoring of PM_2.5_ levels in the predominately African American Ironbound community of Newark, New Jersey [5], Southern California [33], and in school and playground settings in Brooklyn, New York City [48]. Additionally, this citizen science approach can be utilized to assess the urban mosquito abundance in West Baltimore neighborhoods, as trash dumping was more prevalent in this urban region [41]. In these studies, community members played a major role in assisting in collecting air monitoring data and in the clean-up of their neighborhoods.

To address air pollution in South Baltimore, Maryland, The South Baltimore Community Exposure Study was conceived and implemented in response to community members’ concerns regarding their exposure to and the potential negative health impacts of VOCs pollution from industrial and mobile sources. A significant number of heavy industries operate in the South Baltimore communities of Brooklyn, Brooklyn Park, and Curtis Bay. These industrial operations emit 360,479,759 pounds of pollutants into the environment annually. Thus, South Baltimore ranks 12th highest among United States communities for total pollutant releases [49]. Community leaders of the South Baltimore neighborhoods have been fighting for their communities against the increase in industrialization [42].

The Baltimore-Washington Metropolitan area is the most populated urban area, with over 9-million residents, in the Chesapeake Bay watershed. There is a critical need for bottom-up participatory approaches to improve water quality and achieve sustainability goals [50,51], as this area remains plagued by environmental stressors, poor water quality, and hypoxic “dead zones” [52,53]. The two Chesapeake Bay watersheds of focus are Watershed 263 and Watts Branch. These regions are socioeconomically and environmentally diverse and comprised of predominately African American residential populations.

One of the largest shale formations in the United States, known as Marcellus Shale, is found below the surface of several Appalachian states. There is potential to increase fracking operations in the Allegany and Garrett counties of Western Maryland. Research in the social and health sciences on the community impacts of fracking has been conducted in neighboring West Virginia, but the region of Western Maryland has been overlooked by scholars [45]. This region of Western Maryland is demographically like the Marcellus Shale region of West Virginia, with most of the population being rural, older, racially homogenous, and of lower socioeconomic status. Communities in modern Appalachia are choosing to redefine themselves and reject the “hillbilly” and “culture of poverty” stereotypes [54]. With this identity shift, Appalachian communities are forming alliances and social movements to advocate for social and health justice in response to the generations of political, economic, and cultural marginalization they have faced [55]. Community members of Doddridge County, West Virginia took part in focus groups to share their insight and perceptions with researchers about how fracking has impacted their health and environment.

Environmental justice community leaders have established a community-university-government partnership to address local environmental concerns in Charleston, South Carolina through research, community engagement and empowerment, and action [44]. In Graniteville, South Carolina, a small textile town of approximately 7000 residents, a major chlorine spill resulted in nine immediate deaths, 72 hospitalizations for acute health effects from chlorine inhalation, and more than 840 people seeking medical attention. As Graniteville is a medically underserved community, disaster response and recovery were implemented via a community-based participatory research model, comprising of public health officials, researchers, and community organizations [43]. Orange County, North Carolina is a predominately low-income, community of color that borders the Orange County regional landfill. Community-driven research methods are utilized to address these residents’ concerns with the lack of regulated public drinking water and sewer service, storm water management, paved roads and sidewalks, community lighting, curbside solid waste collection, and emergency medical, fire, and police protection services [46].

Dimension 3: Methodologies. In addressing the urban mosquito abundance in West Baltimore neighborhoods, the researchers conducted the Mosquito Stoppers Civic Ecology Practices (MS CEP) project, which aligned their efforts with the process of civic ecology. They engaged with participants through utilization of focus groups that included stakeholders and pre- and post- study surveys regarding their experiences with trash dumping and empowerment to remediate urban trash issues. The researchers and participants met once per month over a 5-month period. The meetings were held at times that allowed participants to engage in the Mosquito Stoppers citizen science outreach activities. Data collection was conducted by reading a brief article on the unequal burden of trash distribution and collection in New York City to 46 West Baltimore residents at two local community events. The reading was followed up by the conduction of interviews, in which the researchers asked the residents the following question: “Do you think the city would respond differently to others making similar complaints (about trash issues) but in different neighborhoods? Why or why not?” The interview responses were transcribed and coded. All codes were aggregated and re-coded, resulting in 100% agreement [41].

Researchers conducted The South Baltimore Community Exposure Study to assess the community residents’ concerns regarding their exposure to air pollution, specifically VOCs, and the associated potential health implications. Community member engagement in air monitoring is becoming an important component for community-based participatory research [56]. The South Baltimore Community Exposure Study began with communication between the researchers and the Concerned Citizens for Better Brooklyn, a well-recognized community organization. The president of the organization held a major role in the design and implementation of the exposure study and the communications strategy. Prior to launching the field study, two meetings were held with the attendance of approximately 50 community residents to discuss how the environmental study would be conducted, to clarify that this was not a health study, and to inform that neighborhood recruitment would take place. The association assisted the researchers in identifying a community member to work on the study as the recruiter. A community advisory committee was established, consisting of four local leaders from each of the South Baltimore communities and one delegate of the Maryland state legislature. Personal and community level monitoring was conducted, along with a risk assessment. A community-wide report of the study results and conclusions was disseminated after all monitoring and data analysis was completed [42].

To understand best management practices, sense of perceived responsibility, barriers, and the future of Green Infrastructure (GI) in two Chesapeake Bay watersheds, researchers utilized a community-based participatory research approach and a transition management framework, to form a community advisory board (CAB), which consisted of a multi-disciplinary team of community residents, local nonprofit organizations, government organizations, and the University of Maryland Extension (UME) [47]. In-depth interviews consisting of open-ended questions were conducted by the researchers. The 42 coded interview transcripts were analyzed.

To understand community concerns with fracking and to investigate its potential health impacts in Doddridge County, West Virginia, researchers utilized the scoping process traditionally employed in a health impact assessment [45]. Focus groups conducted among residents allowed for deeper exploration of their perspectives, which was a useful method, as the residents have a shared history as landowners. To recruit participants, the researchers employed the use of flyers, email blasts, and announcements on the project website. The focus group transcripts were coded using a thematic approach. After finalization of the coding scheme, printed reports were utilized to write a detailed analytical report of the findings.

An environmental justice radar was designed to assist in educating residents of Charleston, South Carolina about the local environmental hazards. Additionally, this website serves as a portal by stakeholders as a tool for visualizing environmental risks for communities of concern [44]. This radar serves as a mapping visualization tool that shows the location of industrial pollution sources. Working with university researchers, community members were directly involved with the design of the tool through feedback sessions at community meetings. The environmental justice radar maps the cumulative and differential burden of social, economic, and health measures organized across five domains: environmental hazards, sociodemographic data, air quality data, soil contamination data, and health data. Community members participate in data sharing by taking pictures of environmental hazards or concerns and posting them to the website. In Washington state, scientists collaborated with residents to develop a cumulative environmental health impacts tool, called the Washington Environmental Health Disparities Map. This visualization tool integrates community voice and the lived experiences of residents in mapping and ranking environmental health disparities throughout the state and provides important information to policymakers, residents, and community organizations for informed decision making [3]. The Drinking Water Tool (DWT) was developed in collaboration between community residents and researchers to provide users with information about drinking water sources, access, and quality in response to draught in California and to assist with informing policy change [57].

A community-based participatory service approach was utilized to bring public health services to victims of a major chlorine spill in Graniteville, South Carolina [43]. After the initial emergency response by the first responders, workers with the state public health agencies (SCDHEC) conducted air monitoring inspections prior to residents being able to reoccupy the area. Community leaders presented concerns to SCDHEC, and along with volunteers, the Graniteville Community Coalition (GCC) was formed. The collaboration assisted residents in recovery and in moving Graniteville toward self-sufficiency after the chlorine disaster. SCDHEC and the GCC collaborated to host community meetings and training workshops where residents could address their concerns and ask questions. Participatory-based training is an important step in the collaborative research process that can encompass cultural competency, interview techniques, and environmental sampling methods [18,58] The community requested that SCDHEC perform further environmental sampling and monitoring to ensure safety and to establish a health tracking registry. To recruit residents into the registry, additional local partnerships with schools, businesses, faith-based organizations, and the University of South Carolina–Aiken were formed. Churches and faith-based organizations have great impact in the success of community-based participatory research methods [59]. Health screenings for residents were conducted at two local churches. Building on the same community-based participatory approach, researchers, and public health workers were able to transition from public health practice to research three years after the chlorine disaster.

To investigate water and sewer disparities and the safety of water supplies among the Rogers-Eubanks community in Orange County, North Carolina, partnerships between the Rogers-Eubanks Neighborhood Association (RENA) and the researchers at the University of North Carolina at Chapel Hill were vital [46]. GIS maps were created with the assistance of community members’ historical knowledge of community boundaries. With a collaboration between community members, local organizations, and researchers, demographic surveys were distributed to households. Drinking, and surface water samples were collected and analyzed.

Dimension 4: Outcomes. Community-based participatory methods are invaluable for building community partnerships and trust among residents and outside institutions. Utilizing community-based participatory research methods, such as resident interviews, focus groups, and Photovoice, provide valuable insights to community residents’ perceptions of their environmental health risks to more effectively direct scientists in their research efforts [4,60,61,62,63]. Community-based participatory research methods also serve as a culturally competent way for scientists to engage with and educate environmentally vulnerable, Spanish-speaking communities [64]. The encouragement of locally based stewardship through citizen science practices improves communities and advocacy, especially among the school-aged youth of communities. [5,41,59,65,66] Citizen science serves as a conduit for adaptive co-management and decision making. Community-based participatory methods can empower communities to collaborate amongst themselves to advocate for their environmental health [46,67]. The formation and inclusion of a community advisory committee/board is invaluable when conducting air quality monitoring studies and risk assessments to gauge community members’ previous knowledge and to assess their expectations of the research [42,68]. Decision-making must involve open dialogue among community members, governing bodies, policymakers, nonprofit organizations, researchers, and environmental professionals to produce constructive outcomes [47]. Community involvement and feedback in the creation of mapping and visualization tools that researchers intend for community residents to utilize is invaluable, as resident engagement is more likely if they are included throughout the entire research process [3,44,46,56,57] Community-based participatory methods are not only valuable in research studies, but also in emergency response situations, and may provide insights for developing disaster prevention strategies [43]. With climate change impacts, such as heat waves, affecting urban populations, community-based participatory research methods should be considered, as these impacts are becoming more of a public health concern for the most vulnerable populations [37,57] However, it is important to consider that the findings from one community-based participatory research study cannot be generalized for all vulnerable communities but may help to extrapolate potential outcomes and provide support for its utilization in other similar communities with environmental concerns or hazards [45].

Dimension 5: Limitations/Challenges. Implementing a community-based participatory approach in a study may present challenges to researchers. Small sample size or the inability to recruit a representative number of residents were noted as limitations [41,45,69]. Another challenge that presented itself when conducting community-based participatory research was the variable attendance of residents at focus groups or events, which presented some difficulty in sustaining individual engagement in some instances [37,41] Additionally, community residents may present with some level of disappointment if the research does not provide the desired or expected outcome between the environmental hazard of focus and the proposed disparate health outcomes, which researchers and stakeholders will need to manage [42]. The tension between science and politics presented itself as a limitation to conducting health and exposure investigations in response to community concerns [42]. A widespread challenge to employing community-based participatory research methods is recognizing the, often historical, distrust residents have of institutions and placing emphasis on building that trust throughout the research process [42,43,45,46,47]

## 4. Future Directions

The E^6^ model, paired with the recently described *Public Health Exposome* framework [1,5,16], provides us with a comprehensive set of tools to investigate and analyze how chemical and non-chemical stressor exposures from the built, natural, physical, and social environments are associated and linked to disparate health outcomes spatially and temporally at an individual, community, and population level. Using the *Public Health Exposome* with its Big Data to Knowledge Analytics, we can extract, normalize, synchronize, and link all survey and health data to estimate risk trajectories in response to the chemical and non-chemical stressors that vulnerable residents are exposed to. The health outcomes of the residents are measured by linking the residents’ electronic medical records to the *Public Health Exposome* database. This is accomplished by utilizing our Public Health Exposome framework with Big Data to Knowledge analytics to analyze the ICD-9 and ICD-10 codes from the residents’ medical records. Other sociodemographic, environmental, behavioral, and personal response outcomes measured from self-report surveys will be extracted, normalized, synchronized, and integrated into the *Public Health Exposome* dataset. Hypothesis testing will be conducted to potentially reveal latent links and associations amongst these variables for residents with disparate health outcomes from exposure to chemical and non-chemical stressors. The ability to pinpoint environmental and socio-demographic variables that are at the core of historic structural inequalities is seminal to our efforts. We have utilized this framework in the high-risk and vulnerable communities of Columbus, Ohio and we have reported on this in the literature [70,71,72,73,74].

## 5. Conclusions

From this systematic review, it can be shown that community-based participatory research has been and continues to be a well-utilized and specialized method of conducting research in vulnerable communities where leaders, researchers, and stakeholders collaborate in the scientific process to ultimately address and mitigate disparate health outcomes. The variations with which community-based participatory research models can be put into practice allow for more impact among diverse populations. Existing models may be nuanced and enhanced to better reach and serve niches within vulnerable communities. The E^6^ model allows for environmental public health stakeholder teams to interface with the populations that are often excluded from public health studies. By forming such community partnerships, we can move towards a more equitable discovery process to ensure that concerns of under-represented minorities and residents of environmental justice communities are addressed in studies. Through implementation of our E^6^ model, we have demonstrated that functional partnerships among academic researchers, stakeholders, community leaders, and residents can be formed to effectively address environmental concerns in Columbus, Ohio.

Lastly, we were also able to introduce the Health Opportunity Index (HOI) as a multivariate tool that can be more efficiently used to identify and understand the interplay of complex social determinants of health (SDH) at the census tract level which influences the ability to achieve optimal health. The derivation of the HOI utilizes the data-reduction technique of principal component analysis to determine the impact of SDH on optimal health at lower census geographies (Figure 4). In the midst of the persistent health disparities and the present COVID-19 syndemic, we demonstrated the potential utility of using 13-input variables to derive a composite metric of health (HOI) score as a means to assist in the identification of the most vulnerable communities during the current syndemic [75]. Ogojiaku et al., (2020) is the article entitled, “The Health Opportunity Index: Understanding the Input to Disparate Health Outcomes in Vulnerable and High-Risk Census Tracts”. We were working to develop a state-wide index that would better identify vulnerable populations across the census tracts in the 88 Ohio counties. The article reports the derivation of this index where we utilized the data-reduction technique of principal component analysis to determine the impact of social determinants of health on the health status of populations at lower census geographies.

While bringing comprehensive primary care and other healthcare services to where vulnerable people live, work, play, and pray is not necessarily a new healthcare delivery model, utilizing our functional, interdisciplinary, community-based research stakeholder team in partnership with residents from vulnerable communities was deemed transformative by the residents of the Near East Side of Columbus, Ohio.

We will pair these models and utilize the *Public Health Exposome* framework with Big Data to Knowledge analytics to provide researchers with a tool kit to better characterize these neighborhoods and quantify the associated links to disparate health outcomes more accurately. Our collective framework offers a more robust citizen science approach to testing hypotheses and generating useful results for vulnerable communities with the goal of achieving The Quadruple Aim of improved individual, community, and population health by: (1) improving community residents experience; (2) improving the health of vulnerable and high-risk populations; (3) lowering the total cost of care to vulnerable residents, thereby leading to improved healthcare provider experience; and (4) identifying latent factors and environmental factors/variables related to chemical and non-chemical stressor exposures associated with disparate health outcomes in environmental justice communities. Ultimately, implementation of our E^6^ model in conjunction with the *Public Health Exposome* framework and Big Data to Knowledge analytics will be scaled and disseminated to like communities throughout the United States [76,77,78,79,80,81].

## Figures and Tables

**Figure 1 ijerph-19-13846-f001:**
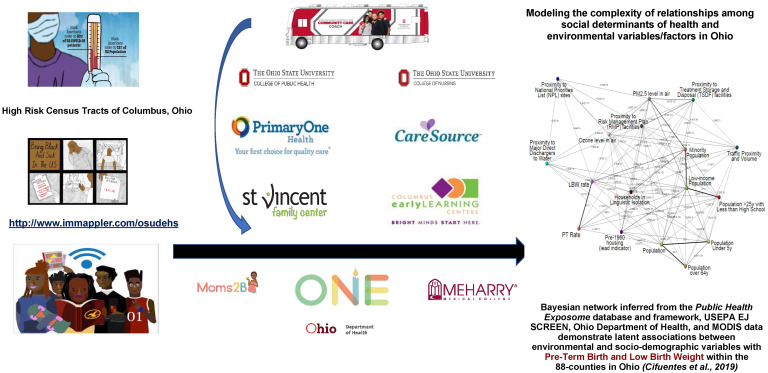
Graphic depiction of our transformative community engagement model that is referred to as E^6^, Enriching Environmental Endeavors via e-Equity, Education and Empowerment. This model utilizes a functional, multidisciplinary, community-based research stakeholder team to educate and improve the quality of life for residents living in environmental justice communities, as well as to foster citizen science among residents and researchers. The E^6^ model prioritizes the perspectives and desires of the residents and is a novel approach to bring effective mitigation strategies directly to where residents live, work, play, and pray (See text for details). [Reproduced and adapted with permission from the corresponding author of Cifuentes et al., 2019 on 18 October 2022] [5].

**Figure 2 ijerph-19-13846-f002:**

The phases of progression and development of the E^6^ framework towards acquisition of a USEPA Science to Achieve Excellence (EPA STAR) grant that addresses concerns of the Near East Side residents (See text for details).

**Figure 3 ijerph-19-13846-f003:**
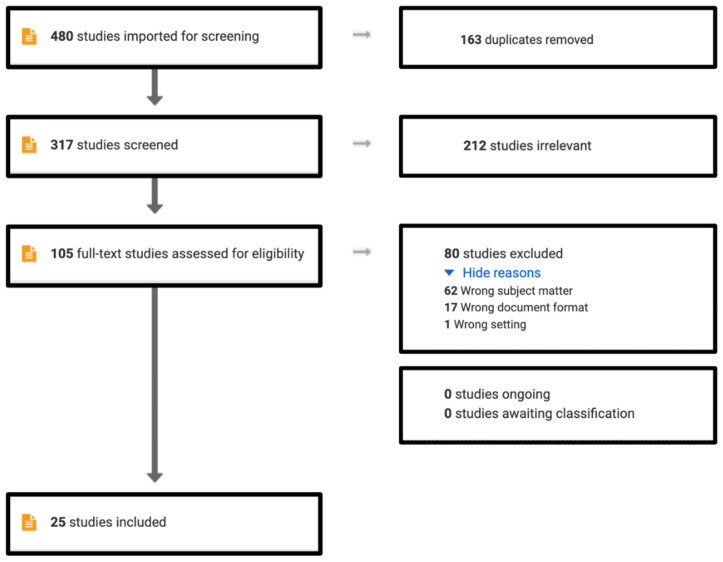
PRISMA diagram for review of literature in PubMed, Scopus, and Embase databases for the present article.

**Figure 4 ijerph-19-13846-f004:**
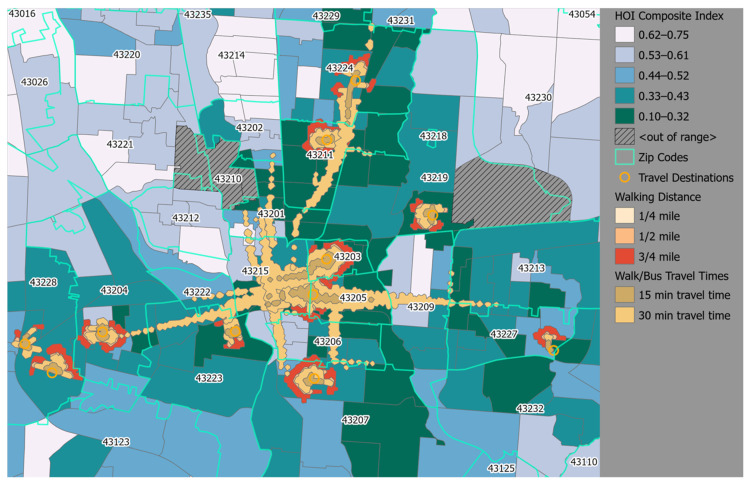
Health Opportunity Index (HOI) composite map of the vulnerable (dark green) census tracts in metropolitan, urban Columbus, Ohio. The census tracts are color coordinated based on what quintile their HOI composite score falls in. In this analysis, the lower HOI composite scores fall in Quintile 1, while the higher HOI scores fall in Quintile 5. [Reproduced and adapted with permission from the corresponding author of Ogojiaku et al., 2020 on 18 October 2022] [75].

**Table 1 ijerph-19-13846-t001:** Summary of socio-demographic and health indicators for the Near East Side by zip-code vs. Franklin County vs. Columbus City vs. the State of Ohio (See Cifuentes et al., 2019 [5] for details). [Reproduced and adapted with permission from the corresponding author of Cifuentes et al., 2019 on 18 October 2022].

Measure	43203	43205	Franklin County	Columbus	Ohio
Population	8108 ^1^	12,272 ^1^	1,163,481 ^1^	787,324 ^1^	11,536,503 ^1^
	8415 ± 1006 ^3^	12,590 ± 1054 ^3^	1,323,807 ^2^	905,748 ^2^	11,799,448 ^2^
Percent African American	62 ± 10 ^3^	49.5 ± 7 ^3^	21.2 ^1^	27.9 ^1^	12.2 ^1^
			22.5 ^2^	28.6 ^2^	12.5 ^2^
Percent White	29.9 ± 7 ^3^	41.1 ± 6 ^3^	69.2 ^1^	61.5 ^1^	82.7 ^1^
			60.6 ^2^	53.2 ^2^	77 ^2^
Educational Attainment for Population 25+ (Percent)					
Less than high school	11.5 ± 3.1 ^3^	12.5 ± 2.8 ^3^	8.2 ± 0.7 ^3^	9.2 ± 0.9 ^3^	8.3 ± 0.2 ^3^
High school +	88.5 ± 3.1 ^3^	87.5 ± 2.8 ^3^	91.8 ± 0.7 ^3^	90.8 ± 0.9 ^3^	91.7 ± 0.2 ^3^
Bachelor’s degree +	23 ± 4.9 ^3^	32.3 ± 4.7 ^3^	41.8 ± 1.2 ^3^	38.4 ± 1.6 ^3^	30.7 ± 0.3 ^3^
Graduate/professional degree +	8.9 ± 4.2 ^3^	11.5 ± 2.9 ^3^	16.1 ± 0.8 ^3^	13.4 ± 0.9 ^3^	11.8 ± 0.2 ^3^
Employment, Income, Poverty, Healthcare Coverage					
Percent of unemployed among individuals > 16 years in civilian labor force	13.8 ± 5.6 ^3^	8.6 ± 2.3 ^3^	4.8 ± 0.2 ^3^	5.4 ± 0.3 ^3^	5.3 ± 0.1 ^3^
Median Household income (in 2020 U.S. dollars)	36,418 ± 5440 ^3^	40,559 ± 3617 ^3^	62,352 ± 706 ^3^	54,902 ± 835 ^3^	58,116 ± 228 ^3^
Percent of families below the poverty level (for which poverty is determined)	35.9 ± 6.9 ^3^	31.2 ± 5.4 ^3^	15.1 ± 0.4 ^3^	19.1 ± 0.6 ^3^	13.6 ± 0.2 ^3^
Percent of insured to health care (civilian non-institutionalized population)	91.8 ± 2.4 ^3^	91 ± 2.3 ^3^	92.3 ± 0.3 ^3^	90.8 ± 0.4 ^3^	93.8 ± 0.1 ^3^
Chronic Health Outcomes (crude prevalence)					
Prevalence of diabetes	20% (19.4,20.5) ^4^	17% (16.5, 17.5) ^4^	10.7 (10.1, 11.4) ^4^	12.4 (8.0–16.9) ^4^	11.8 (11.0, 12.7) ^4^
Prevalence of obesity	48.5% (47.9, 49.2) ^4^	45.9% (45.3, 46.5) ^4^	35.6 (34.4, 36.7) ^4^	37.6 (36.5, 36.8) ^4^	37.6 (36.0, 39.1) ^4^
Prevalence of current asthma (adults)	13.4% (12.9, 13.8) ^4^	12.2% (11.9, 12.5) ^4^	10 (9.5–10.5) ^4^	10.8 (10.7, 10.8) ^4^	10.3 (9.7, 11.0) ^4^
Prevalence of Smoking	28.8% (27.1, 30.6) ^4^	26.7% (25.2, 28.3) ^4^	20.5 (17.5, 23.5) ^4^	22.8 (22.6, 23.0) ^4^	23.3 (20.0, 26.4) ^4^
Prevalence of Depression	19.9% (19.3, 20.4) ^4^	19.7% (19.4, 20.2) ^4^	20.7 (19.6, 21.7) ^4^	21.9 (21.8, 22.0) ^4^	22.5 (21.2, 23.8) ^4^
Adverse Pregnancy Outcomes					
Total Number of Births	824 ^5^	902 ^5^	38,053 ^6^	48,086 ^5^	674,202 ^7^
Percent Low Birth Weight	15.7 ^5^	14.5 ^5^	9.0 ^6^	10.5 ^5^	8.6 ^7^
Percent of Preterm Births	14.8 ^5^	15.0 ^5^	10.5 ^6^	11.6 ^5^	10.4 ^7^
Percent of Teenage Mothers	9.6 ^5^	7.9 ^5^	1.5 ^6^	6.2 ^5^	5.4 ^7^
Infant Mortality Rate (2010–2014)	13.3 ^5^	10.0 ^5^	8.4 ^5^	9.3 ^5^	7.5 ^7^
Leading Causes of Death (cases; ADR (95% CI))					
Heart Disease	61; 270.9 ^8^	95; 325.0 ^8^	176; 205.28 ^8^	2060; 165.6 ^9^	29,159; 188.8 ^9^
Cancer	47; 195.8 ^8^	67; 226.8 ^8^	172; 205.6 ^8^	1934; 149.7 ^9^	25,166; 163.0 ^9^
Stroke	18; 79.2 ^8^	17; 69.8 ^8^	51; 56.9 ^8^	515; 43.1 ^9^	6504; 42.1 ^9^
Chronic Lower Respiratory Disease	**	15; 48.3 ^8^	41; 48.7 ^8^	559; 44.8 ^9^	7168; 45.9 ^9^
Diabetes	13; 96.5 ^8^	18; 58.9 ^8^	32; 38.0 ^8^	305; 23.8 ^9^	3876; 25.4 ^9^
Accident/Unintentional Injury	18; 81.5 ^8^	35; 92.3 ^8^	54; 61.7 ^8^	918; 70.1 ^9^	8.291; 67.7 ^9^
Homicide	**	12; 27.3 ^8^	19; 21.3 ^8^	115; 8.5 ^9^	726; 6.6 ^9^
Suicide	**	**	14; 16.1 ^8^	153; 11.4 ^9^	1809; 15.2 ^9^

^1^ 2010 Census; ^2^ 2020 Census. ^3^ 2020 American Community Survey; ^4^ CDC PLACES, 2021 release, (2018-19 BRFSS, 2010 Census, 2015-19 ACS): https://chronicdata.cdc.gov/500-Cities-Places/PLACES-ZCTA-Data-GIS-Friendly-Format-2021-release/kee5-23sr/data (accessed on 26 September 2022); ^5^ Ohio Department of Health Vital Statistics, Analysis by Office of Epidemiology, Columbus Public Health (2016–2020); ^6^ Columbus Public Health–CPH Epi Program. ^7^ Ohio Department of Health Vital Statistics (2016–2020); ^8^ Ohio Department of Health Vital Statistics, Analysis by Office of Epidemiology, Columbus Public Health (2015–2017); ^9^ Ohio Department of Health Vital Statistics (2019); ** Data not available due to small numbers.

## Data Availability

Aggregate data from this study is readily available by contacting the corresponding author. All data is stored within a customized, secure cloud environment in a data lake maintained by The Ohio State University and Meharry Medical College.

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
