# Peer review of "A Framework for Interfacing and Partnering with Environmental Justice Communities as a Prelude to Human Health and Hazard Identification in the Vulnerable Census Tracts of Columbus, Ohio"

_ijerph, 2022, doi:10.3390/ijerph192113846_

Round 1

Reviewer 1 Report

Dear, Prof. Hood,

This review manuscript is interesting and important because of presenting and summarizing the development of community engagement framework to identify potential environmental hazards, and building positively the systems associated with residents in vulnerable communities with disparity related to the substantial wealth gap. This review itself seems scientifically sound, and acceptable for International Journal of Environmental Research and Public Health with the research approach and the social importance, but I think that this manuscript is necessary to be revised in minor scale in rhetorical points. I described several comments for the points I care with attached file.

 Manuscript (ID: IJERPH-1965053)

General comments:

  This review manuscript is interesting and important because of presenting and summarizing the development of community engagement framework to identify potential environmental hazards, and building positively the systems associated with residents in vulnerable communities with disparity related to the substantial wealth gap. This review itself seems scientifically sound, and acceptable for International Journal of Environmental Research and Public Health with the research approach and the social importance, but I think that this manuscript is necessary to be revised in minor scale in rhetorical points.

Specific comments:

Page 1

Abstract:

Line 22 and 23: It is better to describe this sentence as follows: ~ at birth, has a census tract wealth gap (27-years disparity), and ~

Line 23: The reference number (1) is not necessary in abstract.

Line 33: I think that “that” should be removed.

Keywords:

Line 36: Community-based participatory research; CBPR

→ Community-based participatory research (CBPR)

Introduction:

Line 45: Please confirm and unify the annotation for the area and the community; near the East Side (Line 45), Near East Side (Line 98, 142, 146), the Near East side neighborhoods (Line 111), Near East Side neighborhoods (Line 177), the near east side (Line 564).

Page 2

Line 71: The period should be removed before the parethesis.

Page 3

Line 104: Please insert a blank between “black” and “and”.

Line 112: The period should be removed before the parenthesis. Please describe the reference number 6.

Materials and Methods:

Page 4

Line 143: The period should be removed before the parenthesis.

Page 5

Line 192: The formal name (Public Participatory Geographical Information System) is not necessary because the formal name has been already described at Line 48 and 49.

Results and Discussion from Community Meetings:

Line 215: The formal name for NAACP should be described.

Page 7

Line 260: It is better to replace “their” with “its”.

Page 8

Line 314: The description with “PM2.5” is recommended instead of “PM2.5” because the abbreviation has been already described at Line 87.

Line 324: The formal name (volatile organic compounds) is not necessary because the formal name has been already described at Line 305 and 306.

Page 9

Line 384: The formal name (volatile organic compounds) is not necessary because the formal name has been already described at Line 305 and 306.

Future Directions

Page 12

Line 515: Please confirm and unify the annotation for the analysis; Big Data to Knowledge Analytics (Line 515), Big Data to Knowledge analysis (Line 520), Big Data to Knowledge analytics (Line 566, 577).

Conclusions

Line 535: Please confirm the blank size between sentences.

Line 559: The period should be removed before the parenthesis. Please describe the reference number 75.

Author Response

Dear Reviewer,

Thank you for your thoughtful comments. Please find the responses to your comments below.

Concerns: Abstract:

  • Line 22 and 23: It is better to describe this sentence as follows: ~ at birth, has a census tract wealth gap (27-years disparity), and ~
  • Line 23: The reference number (1) is not necessary in abstract.
  • Line 33: I think that “that” should be removed.

Responses:

  • Line 22 and 23: This sentence has been edited in the manuscript to read as, “However, it ranks as the second worst life-expectancy at birth, has a census tract wealth gap (27-years disparity), and one of the higher infant mortality rates in the country.”
  • Line 23: Reference number (1) has been removed from the manuscript.
  • Line 33: The word “that” has been removed in the manuscript and the sentence now reads as “The refined framework uses a blended version of traditional community-based participatory research (CBPR) models and is referred to as E6, Enhancing Environmental Endeavors via e-Equity, Education, and Empowerment.”

 Concern:

Keywords:

  • Line 36: Community-based participatory research; CBPR

→ Community-based participatory research (CBPR)

Response:

  • Line 36: “Community-based participatory research; CBPR” has been changed to “Community-based participatory research (CBPR)” in the manuscript.

Concerns:

Introduction:

  • Line 45: Please confirm and unify the annotation for the area and the community; near the East Side (Line 45), Near East Side (Line 98, 142, 146), the Near East side neighborhoods (Line 111), Near East Side neighborhoods (Line 177), the near east side (Line 564).
  • Line 71: The period should be removed before the parenthesis
  • Line 104: Please insert a blank between “black” and “and”.
  • Line 112: The period should be removed before the parenthesis. Please describe the reference number 6.

Responses:

  • Line 45, 98, 111, 142, 146, 177, 564: The annotation for the area and the community has been confirmed, unified, and edited in the manuscript to read as “the Near East Side”.
  • Line 71: The period before the parenthesis has been deleted in the manuscript.
  • Line 104: A space has been inserted and the sentence in the manuscript now reads as “Exposures to chemical and non-chemical stressors are thought to adversely impact allostatic load and contribute to the disparity observed among black and white COVID-19 mortality rates.13
  • Line 112: The period has been removed before the parenthesis. Reference number 5 (previously reference number 6) is the publication that served as the scientific premise for our continued involvement with the Near East Side. This publication sought to understand the association of several environmental and socio-demographic variables with adverse pregnancy outcomes across the 88 counties in Ohio by modelling an African American woman’s cohort in Ohio. The adverse outcomes queried were pre-term birth (PTB) and low birth weight (LBW) as a proxy for infant mortality.

(This language has been added to Lines 140-145 in the manuscript.)

 Concerns:

Materials and Methods:

  • Line 143: The period should be removed before the parenthesis.
  • Line 192: The formal name (Public Participatory Geographical Information System) is not necessary because the formal name has been already described at Line 48 and 49.

Responses:

  • Line 143: The period before the parenthesis has been removed in the manuscript.
  • Line 192: “Public Participatory Geographical Information System” has been deleted and replaced with PPGIS in the manuscript.

 Concerns: 

Results and Discussion from Community Meetings:

  • Line 215: The formal name for NAACP should be described.
  • Line 260: It is better to replace “their” with “its”.
  • Line 314: The description with “PM5” is recommended instead of “PM2.5” because the abbreviation has been already described at Line 87.
  • Line 324: The formal name (volatile organic compounds) is not necessary because the formal name has been already described at Line 305 and 306.
  • Line 384: The formal name (volatile organic compounds) is not necessary because the formal name has been already described at Line 305 and 306.

Responses:

  • Line 215: The “NAACP” has been replaced with “the National Association for the Advancement of Colored People (NAACP)” in the manuscript.
  • Line 260: “Their” has kept in the manuscript, as the sentence is referencing the residents.
  • Line 314: “5” has been replaced with “PM2.5” in the manuscript.
  • Line 324: “Volatile organic compounds” has been replaced with “VOCs” in the manuscript.
  • Line 384:Volatile organic compounds” has been replaced with “VOCs” in the manuscript.

 Concern:

Future Directions

  • Line 515: Please confirm and unify the annotation for the analysis; Big Data to Knowledge Analytics (Line 515), Big Data to Knowledge analysis (Line 520), Big Data to Knowledge analytics (Line 566, 577).

Response:

  • Line 515: The annotation has been confirmed and unified in the manuscript as “Big Data to Knowledge analytics”.

 Concerns:

Conclusions:

  • Line 535: Please confirm the blank size between sentences.
  • Line 559: The period should be removed before the parenthesis. Please describe the reference number 75.

Responses:

  • Line 535: The blank size has been adjusted in the manuscript.

Line 559: The period before the parenthesis has been removed in the manuscript. The description of reference number 75 has been included in the article: “Ogojiaku et al., 2020 is the article entitled “The Health Opportunity Index: Understanding the Input to Disparate Health Outcomes in Vulnerable and High-Risk Census Tracts”. We were working to develop a state-wide index that would better identify vulnerable populations across the census tracts in the 88 Ohio counties. The article reports the derivation of this index where we utilized the data-reduction technique of principal component analysis to determine the impact of social determinants of health on the health status of populations at lower census geographies.” (This language has been added to Lines 677-682 in the manuscript.)

Reviewer 2 Report

As a suggestion I would like to tell the authors the following things:

1. make the purpose of the work more explicit in the introduction

2. describe more accurately the model you intend to use (E model)

3. describe in more detail the statistical methodologies that are intended to be used to manage the dates once acquired

There are also numerous formatting errors in the text

Line 75 th should be the

the E6 model now  is well explained in the introduction as I suggest.

Line 141:  the number 6 has to be an apice

Line 224-227

“Dr. Hood lead discussions on how these issues are studied in a public  health research setting. He explained the Public Health Exposome framework, Bayesian network analysis, and the characterization of how exposure to various chemical and non-chemical stressors might influence residents’ susceptibility to disease. “

this concept should be explained in more detail, especially how the data were used from a statistical point of view

Line 314: wrong format

Dimencion 6

It would be appropriate to provide some other detail between the ratio of official monitoring data and the data collected from the population in order to bring out the difference between actual risk and risk perceived by the population.

The agreement between the official data and those collected from the population could guarantee the effective effectiveness of the instrumentation used for participatory monitoring

Community- 485 based participatory methods are not only extremely valuable for research studies, but also 486 in emergency response situations, by providing insights for developing disaster preven- 487 tion strategies.

can they lead to false alarms or the incorrect interpretation of the results can lead to an exaggerated perception of risk?

Health Opportunity Index

This concept should be explored and explained more thoroughly

Author Response

Dear Reviewer,

Thank you for your thoughtful comments. Please see the responses to your comments below.

Concern: Make the purpose of the work more explicit in the introduction.

Response: The following language has been included in Lines 150-153 of the manuscript: Conclusion of the community meeting series coincided with the nation going into lockdown due to the COVID-19 syndemic. As a result, for the remainder of 2020, 2021, and the first quarter of 2022, all materials and data were organized and analyzed. A grant proposal was then prepared and submitted to query many of the questions and concerns presented by the community residents.”

Concern: Describe more accurately the model you intend to use (E model).

Response: The following language has been included in Line 41-58 of the manuscript: “The purpose of the E6 model can be utilized to bring comprehensive primary care and health care services to where people live, work, play and pray. Over the past 50-years, this has proved to be easier said than done in the vulnerable census tracts of the United States. Our efforts to formalize and create a template for creation of a functional, interdisciplinary, community-based research stakeholder team in true partnership with residents is transformative and will positively impact individual, community, and population health. This model can assist with informing the decision-making processes related to resource allocation for high-risk and vulnerable communities by local and state environmental public health policy officials.”

Concern: Describe in more detail the statistical methodologies that are intended to be used to manage the dates once acquired.

Response: Thank you for your comment. However, for us to better address this, can you please provide a more granular articulation of the comment?

Concern: There are also numerous formatting errors in the text.

Response: Formatting edits and corrections have been addressed in the manuscript.

Concern: Line 75 th should be the

Response: “th” has been corrected to “the” in the manuscript.

Concern: Line 141:  the number 6 has to be an apice

Response: The manuscript has been updated to read “E6”, rather than “E6”.

Concern: Line 224-227: “Dr. Hood lead discussions on how these issues are studied in a public health research setting. He explained the Public Health Exposome framework, Bayesian network analysis, and the characterization of how exposure to various chemical and non-chemical stressors might influence residents’ susceptibility to disease.” This concept should be explained in more detail, especially how the data were used from a statistical point of view.

Response: The following language has been added to Line 323-335 in the manuscript: “Associations between socio-demographic and environmental variables and an adverse health or otherwise negative outcome were analyzed, as reviewed in Cifuentes et al., (2019). These associations accounted for 32.85% density and an average degree of 9.2. Post hoc values of arrows (associations) were plotted as p-values based on linear conditional correlation and line widths were highest for the lowest p-values. Automatic visualization accounted for the relative value of the links, which was obtained by transforming p-values by log-transformation and normalization/truncation from 5 to 1 by a mapping algorithm. This was followed by an energy-based algorithm, available in Pajek software, which located more connected nodes in the center of the graph. The p-values for each association were calculated to derive an adjacency matrix representing significant associations between pairs of environmental and/or socio-demographic variables that are controlled by all remaining variables. The weighted adjacency matrix of the resulting Bayesian network model contained all significant p-values (< 0.05) for each link.5

Concern: Line 314: wrong format

Response: The formatting has been addressed and corrected in the manuscript.

Concern: Dimencion 6

Response: Thank you for your comment. However, for us to better address this, can you please provide a more granular articulation of the comment?

Concern: It would be appropriate to provide some other detail between the ratio of official monitoring data and the data collected from the population in order to bring out the difference between actual risk and risk perceived by the population.

Response: Thank you for your comment. However, for us to better address this, can you please provide a more granular articulation of the comment?

Concern: The agreement between the official data and those collected from the population could guarantee the effective effectiveness of the instrumentation used for participatory monitoring.

Response: Thank you for your comment. However, for us to better address this, can you please provide a more granular articulation of the comment?

Concern: Community- 485 based participatory methods are not only extremely valuable for research studies, but also 486 in emergency response situations, by providing insights for developing disaster preven- 487 tion strategies. Can they lead to false alarms or the incorrect interpretation of the results can lead to an exaggerated perception of risk?

Response: I have updated the language in Line 600 in the manuscript.

Concern: Health Opportunity Index. This concept should be explored and explained more thoroughly.

Response: The following language has been added to Lines 677-682 in the manuscript: “Ogojiaku et al., (2020) is the article entitled “The Health Opportunity Index: Understanding the Input to Disparate Health Outcomes in Vulnerable and High-Risk Census Tracts”. We were working to develop a state-wide index that would better identify vulnerable populations across the census tracts in the 88 Ohio counties. The article reports the derivation of this index where we utilized the data-reduction technique of principal component analysis to determine the impact of social determinants of health on the health status of populations at lower census geographies.”